# Effect of Body Composition Change during Neoadjuvant Chemotherapy for Esophageal Squamous Cell Carcinoma

**DOI:** 10.3390/jcm11030508

**Published:** 2022-01-20

**Authors:** Sachiyo Onishi, Masahiro Tajika, Tsutomu Tanaka, Keisaku Yamada, Tomoyasu Kamiya, Tetsuya Abe, Eiji Higaki, Hironori Fujieda, Takuya Nagao, Yoshitaka Inaba, Kei Muro, Masahito Shimizu, Yasumasa Niwa

**Affiliations:** 1Department of Endoscopy, Aichi Cancer Center Hospital, 1-1 Kanokoden, Chikusa-ku, Nagoya 464-8681, Japan; soonishi@aichi-cc.jp (S.O.); tstanaka@aichi-cc.jp (T.T.); k.yamada@aichi-cc.jp (K.Y.); t.kamiya@aichi-cc.jp (T.K.); yniwa@aichi-cc.jp (Y.N.); 2Department of Gastroenterological Surgery, Aichi Cancer Center Hospital, 1-1 Kanokoden, Chikusa-ku, Nagoya 464-8681, Japan; tabe@aichi-cc.jp (T.A.); ehigaki@aichi-cc.jp (E.H.); h.fujieda@aichi-cc.jp (H.F.); t.nagao@aichi-cc.jp (T.N.); 3Department of Diagnostic and Interventional Radiology, Aichi Cancer Center Hospital, 1-1 Kanokoden, Chikusa-ku, Nagoya 464-8681, Japan; 105824@aichi-cc.jp; 4Department of Clinical Oncology, Aichi Cancer Center Hospital, 1-1 Kanokoden, Chikusa-ku, Nagoya 464-8681, Japan; kmuro@aichi-cc.jp; 5Department of Gastroenterology/Internal Medicine, Gifu University Graduate School of Medicine, 1-1 Yanagido, Gifu 501-1194, Japan; shimim-gif@umin.ac.jp

**Keywords:** esophageal squamous cell carcinoma, neoadjuvant chemotherapy, sarcopenia

## Abstract

Effects of changes in body composition during neoadjuvant chemotherapy (NAC) on perioperative complications and prognosis are unknown in patients with esophageal squamous cell carcinoma (ESCC). A total of 175 patients who underwent surgery for ESCC in our hospital between 2016 and 2019 were examined. The psoas muscle index (PMI) was calculated from the total psoas muscle area, and the visceral fat mass (VFM) at the umbilical level was measured. We defined body composition change (BCC) group as those with increased VFM of ≥ 3% and decreased PMI of ≥ 3% during NAC. Sarcopenia (S) was defined as PMI < 5.89 (male) and <4.06 (female). Nutritional assessment using the Subjective Global Assessment tool was performed upon admission. The percentages of BCC group, pre-NAC S, and post-NAC S was 32.5%, 79.4%, and 80.0%, respectively. BCC group had significantly more postoperative complications (*p* < 0.01) and longer hospital stays (*p* = 0.03) than groups pre-NAC S and post-NAC S. Overall survival (OS) analysis using the Cox hazard model showed that stage III (*p* < 0.01) and post-NAC S (*p* = 0.03) were poor prognostic factors. Changes in body composition during NAC affected perioperative complications and prognosis of patients with ESCC.

## 1. Introduction

In 2018, esophageal cancer was reported to have the seventh-highest morbidity rate and the sixth-highest mortality rate among all malignancies; esophageal cancer is more common in men, with a high prevalence in those between 60–70 years old [1]. More than 90% of esophageal cancers are squamous cell carcinomas [2]. The incidence is particularly high in East Asia, Central Asia, East Africa, and the South African, and the same is true in Japan. The risk factors include alcohol consumption and smoking [3]. In the case of advanced esophageal cancer, it is common for the patient to be malnourished at the time of diagnosis since the passage obstruction is caused by the tumor. In recent years, nutrition and body composition reportedly affect the overall survival (OS) rate of esophageal cancer patients [4]. Sarcopenia is one of the most common abnormalities in body composition and is characterized by the progressive and systemic loss of skeletal muscle mass and strength, leading to disability, decreased quality of life, death, among other adverse outcomes [5]. In 2018, the diagnostic criteria for sarcopenia were revised to include muscle strength as the main parameter, but a loss of skeletal muscle mass was still required for a definitive diagnosis [6]. We have previously reported that sarcopenia is a poor prognostic factor in patients with unresectable advanced esophageal cancer [7]. Moreover, in resectable advanced esophageal cancer, sarcopenia affects long-term prognosis and increases the rate of perioperative complications [8,9].

Based on the results of the Japan Clinical Oncology Group (JCOG) study [10], neoadjuvant chemotherapy (NAC) is strongly recommended for clinical stage II or III thoracic esophageal cancer and is positioned as the standard treatment. NAC is limited to two or three sessions, but the side effects of chemotherapy, such as loss of appetite and general fatigue, often result in reduced food intake and decreased activities of daily living (ADL). Patients with esophageal cancer are often malnourished at diagnosis owing to primary tumor stenosis, and neoadjuvant chemotherapy (NAC) may worsen their nutritional status. While there is a report of significant decreases in skeletal muscle index (SMI) and fat mass during NAC [11], there is also a report of no decrease in fat mass [12] due to nutritional management during NAC. Fat mass, especially visceral fat mass, is associated with insulin resistance and increased mortality risk [12,13,14]. In colorectal cancer, visceral and subcutaneous fat masses are prognostic factors [15,16]. Therefore, an increase in fat mass is likely to have a negative impact, even under nutritional management.

In patients with stage II or III esophageal cancer, there are reports that pre-operative sarcopenia is associated with more perioperative complications [9,17] and poorer prognoses [4,8,18,19,20]. There have been few reports on the effects of changes in the crossover between SMI and fat mass, especially for NAC in esophageal cancer. This is probably due to anorexia associated with NAC and the symptoms associated with esophageal strictures often prevent adequate nutrition. In recent years, however, there have been cases in which the deterioration of nutritional status does not progress since nutritional management is performed by enteral or parenteral nutrition along with NAC, and in some cases, NAC may improve oral intake. However, it is unclear whether the increased fat mass is sufficient even if skeletal muscle mass is decreased for management during NAC. The study investigated the effect of NAC to the body composition in stage II and III resectable esophageal squamous cell carcinoma (ESCC) patients.

## 2. Materials and Methods

### 2.1. Study Population

This was a retrospective cohort study of 201 patients with stage II or III thoracic esophageal cancer who underwent surgery at the Aichi Cancer Center between January 2016 and December 2019. Among 201 patients, we excluded 13 for whom NAC was not performed and 13 for whom computed tomography (CT) was not performed before and after NAC, and finally, 175 patients were included. In all cases, neck and pelvic CT were performed before NAC for diagnostic purposes before treatment and after NAC to determine NAC’s therapeutic effect. The staging of esophageal cancer in this study was based on the 11th edition of the Japanese Classification of Esophageal Cancer published by the Japanese Esophageal Society in 2015 [21]. Our study was approved by the Ethics Review Committee of the Aichi Cancer Center (IR031079 2021/09/15) and was conducted per the 1975 Declaration of Helsinki, as revised in 1983.

### 2.2. Neoadjuvant Chemotherapy

In our hospital, the departments of esophageal surgery, clinical oncology, radiation oncology, and endoscopy conduct joint conferences to determine treatment strategies. NAC is based on FP therapy (fluorouracil (5-FU) + cisplatin (CDDP)) or DCF (docetaxel (DOC) + 5-FU + CDDP). The decision of which chemotherapy regimen to use is also discussed and decided at the joint conference. FP therapy consisted of two courses repeated every 3 weeks; 5-FU (800 mg/m^2^) was administered continuously on days 1–5, and CDDP (80 mg/m^2^) was administered only on day 1. DCF therapy was repeated every 3 weeks for three courses. For DCF therapy, 5-FU (700 mg/m^2^) was administered continuously on days 1–5, and DOC (70 mg/m^2^) and CDDP (70 mg/m^2^) were administered only on day 1. Before administration, creatinine clearance was calculated, and the dosage was determined according to each case. In addition, various drugs were reduced and adjusted depending on the toxicity of the previous course.

Before treatment, we checked for hepatitis B virus (HBV) infection. If an HBV infection was observed, the patient was treated with continuous entecavir or tenofovir. If febrile neutropenia (FN) was suspected, the initial evaluation was promptly performed, and antibacterial therapy was initiated. Granulocyte colony-stimulating factor preparations were also administered as prophylactic therapy, especially for elderly or malnourished patients with risk factors for developing FN. All patients underwent surgery at least 3–4 weeks after the last chemotherapy.

### 2.3. Evaluation of Chemotherapy-Related Toxicities and Postoperative Complications

The Common Terminology Criteria for Adverse Events version 5.0 was used to evaluate the adverse events. The Clavien-Dindo classification [22] was used to evaluate postoperative complications; grade 2 represented a complication of medical therapy, including blood transfusion and central venous feeding; grade 3 represented a complication requiring surgical treatment, endoscopic treatment, or interventional radiology (IVR); and grade 4 was defined as a life-threatening complication with ICU management. Treatment response was evaluated according to the Revised Response Evaluation Criteria in Solid Tumors guidelines.

### 2.4. Measurement and Definitions of Body Composition

CT was used to assess the patients’ body composition. The patients in this study underwent CT scans before NAC for diagnostic purposes and after NAC to determine the effects of chemotherapy. In this study, we measured skeletal muscle mass and visceral fat mass using contrast-enhanced CT before and after NAC. The CT images before and after NAC were used to measure the cross-sectional area of the left and right psoas muscles at the third level of the lumbar spine by manual tracing. The cross-sectional area was normalized by height to calculate Psoas Muscle Index (PMI; cm^2^/m^2^). The use of PMI in this study was based on a report by Hamaguchi et al. [23], stating that SMI using bioelectric impedance analysis (BIA) and PMI showed a positive correlation. They proposed a cutoff value for PMI of 6.36 cm^2^/m^2^ for men and 3.92 cm^2^/m^2^ for women, and this cutoff value is also adopted in the Japan Society of Hepatology guidelines for sarcopenia in liver disease (1st edition) [24] proposed by the Japanese Society of Hepatology in 2016. In this study, we used this as the cutoff value for the PMI. We measured the visceral fat mass at the umbilical level on the same CT scan. Visceral fat was quantified in the range of −200—50 HU. Skeletal muscle mass and visceral fat mass were analyzed using the Volume Analyzer Synapse VINCENT 3 image analysis system (Fujifilm Medical, Tokyo, Japan). The percentage change in PMI and visceral fat mass before and after NAC was calculated as follows:%△PMI = (Post-NAC PMI − Pre-NAC PMI)/Pre-NAC PMI × 100(1)
%△Visceral fat = (visceral fat after NAC − visceral fat before NAC)/visceral fat before NAC × 100(2)

In this study, a change of %△PMI and %△Visceral fat ≥ 3% was considered positive, and the group with decreased skeletal muscle mass despite increased visceral fat mass was defined as body composition change (BCC) group.

### 2.5. Patient Data

All patients’ clinical data were collected from a database maintained at our hospital. Height, weight, and body mass index (BMI) were measured at the first visit to our hospital. Laboratory data were obtained from the first visit and the first surgical consultation after NAC. The prognostic nutrition index (PNI) proposed by Onodera et al. [25] had been associated with the outcome of various malignancies, and the PNI was calculated using laboratory data: PNI = (10 × albumin) + (0.005 × lymphocyte count)(3)

The patients’ general condition was assessed using the American Society of Anesthesiologists-physical status (ASA-PS). Mortality data were collected from medical records or by contacting the patient’s family physician or the hospital to which the patient was transferred. Mortality data were collected from the first admission to our hospital until the date of death or the sensor date of the study.

### 2.6. Nutritional Screening

When patients are admitted to our hospital, nutritionists and nurses dedicated to the Nutrition Support Team (NST) assess their nutritional status using the Subjective Global Assessment (SGA), a nutritional screening tool. Based on the SGA questionnaire, patients were classified into three groups: A, well-nourished; B, mild-to-moderate malnutrition; and C, severely malnourished. In addition, NST intervention or nutritional guidance will be provided on a case-by-case basis upon patient request.

### 2.7. Statistical Analysis

Statistical analysis was performed using JMP version 9.0.2 software (SAS Institute, Cary, NC, USA). Continuous variables are expressed as median, mean, and range, and the differences between medians were analyzed nonparametrically using the Mann-Whitney U test. Categorical variables are indicated by the number of patients, and between-group distribution differences were analyzed using the chi-square test. Survival curves were estimated using the Kaplan-Meier method, and comparisons were made using the log-rank test. Univariate or multivariate Cox regression analyses were performed to examine the predictors of overall survival (OS) in the patients. *p* < 0.05 was considered statistically significant.

## 3. Results

### 3.1. Background Factors

The average age of patients who underwent surgery after NAC was 66.2 years old. There were 139 males and 36 females, and their clinical stages were cStages II/III (64/111). NAC was performed in FP/DCF = 122/53 of patients. Before NAC, the mean albumin level was 4.1 g/dL, and the mean PNI was 49.3; after NAC, the mean albumin level was 3.9 g/dL, and the mean PNI was 46.8. The mean prealbumin level (known as rapid turnover protein with a half-life of approximately 48 h) was 26 mg/dL, which was on the low end of the reference value (22–40 mg/dL). Eight patients were managed with enteral nutrition, six with intravenous nutrition, and eleven with oral nutritional supplements, but only seventeen (9.7%) received NST care (Table 1).

### 3.2. Body Composition Changes before and after NAC

Pre- and post-NAC changes in PMI and visceral fat mass are shown in Table 2. There were 32 patients (18.3%) with increased PMI and 80 (45.7%) with increased visceral fat mass. Eighty-six (49.1%) patients had decreased PMI and visceral fat mass and 57 (32.6%) had increased visceral fat despite decreased PMI. We focused on this group and defined them as group A and the other patients as Other.

### 3.3. Characteristics of BCC Group

In terms of sex, there were significantly higher percentages of women in BCC group (*p* = 0.03), but there were no significant between-group differences in age, chronic diseases, or performance status (PS). In addition, the pre-NAC BMI was significantly lower in BCC group (*p* < 0.01; Table 3), but there was no significant difference in the pre- and post-NAC sarcopenia rate, albumin (Alb), or PNI (Table 4). We also found no significant difference in tumor factors in the cStage (*p* = 0.72) or NAC regimen (*p* = 0.72).

### 3.4. Perioperative-Related Factors

This study investigated the operative time, length of hospitalization, and perioperative complications. The operative time was not significantly different from that of the comparison group in pre-NAC sarcopenia (*p* = 0.84), post-NAC sarcopenia (*p* = 0.71), and BCC group (*p* = 0.53). There was no significant difference in the length of hospitalization between the respective comparators for pre-NAC and post-NAC sarcopenia. However, BCC group had a significantly longer hospital stay than the other groups (*p* = 0.03). In the study of perioperative complications of grade 3 or higher, there was no significant difference between comparators for pre-NAC and post-NAC sarcopenia. However, BCC group had significantly more perioperative complications than Other (*p* < 0.01), and among perioperative complications in BCC group, Sugical Site Infection (SSI) was significantly more frequent in BCC group (*p* = 0.02) (Table 5).

### 3.5. Prognostic Factors for OS

In the univariate analysis for OS, cStage III and post-NAC sarcopenia were found to be associated, and in the Cox hazard regression model for OS, cStage III and post-NAC sarcopenia were identified as independent poor prognostic factors (cStage III: hazard ratio (HR) 3.94 (1.64–11.58), *p* < 0.01 median survival time not reached, The three-year survival rates for Stage II and III are 81.1% and 71.2%, respectively; post-NAC sarcopenia: HR 2.92 (1.04–12.1), *p* = 0.03 median survival time not reached; the three-year survival rates for post-NAC sarcopenia and non-post-NAC sarcopenia were 75.9% and 92.5%, respectively). (Table 6) (Figure 1 and Figure 2)

## 4. Discussion

Sarcopenia is characterized by the progressive and systemic loss of skeletal muscle mass and strength, leading to adverse outcomes, such as reduced quality of life and mortality [5]. There have been reports that sarcopenia affects the prognosis of various malignancies [26,27,28]. We have also reported that sarcopenia is a poor prognostic factor in patients with unresectable esophageal cancer [7] and that sarcopenic obesity is a poor prognostic factor in patients with resectable esophageal cancer [29]. In locally advanced esophageal adenocarcinoma, Fehrenbach et al. evaluated preoperative and postoperative CT scans and assessed the impact of body composition on the perioperative morbidity and survival after surgery of patients who received NAC or neoadjuvant chemoradiation therapy. They concluded that sarcopenia patients associated with postoperative complications, especially in sarcopenia obesity, were more prone to postoperative pneumonia. Low preoperative muscle mass and its decrease during postoperative period are associated with poor prognosis [28]. However, the effects of NAC only on body composition abnormalities and the prognostic impact of changes during NAC in patients with esophageal cancer remain unclear. It has been reported that androgen deprivation therapy, a prostate cancer treatment, causes body composition changes such as weight, fat gain, and muscle mass loss approximately 3–6 months after the start of treatment due to adverse effects [30,31]. These changes in body composition are known to cause sarcopenic obesity [32], which is associated with an increased risk of lipid disorders, insulin resistance, and cardiovascular events [33].

On the other hand, recent reports indicate that changes in SMI are an independent risk factor for non-cancer mortality in prostate cancer, but changes in obesity are not, and that skeletal muscle mass loss is potentially independent of changes in body weight or fat mass and is associated with mortality in prostate cancer patients [34]. We reported the presence of sarcopenic obesity in patients with thoracic esophageal cancer receiving neoadjuvant chemotherapy and its impact on prognosis [29]. Based on previous reports [32], it can be predicted that there is a decrease in skeletal muscle mass and an increase in fat mass in the background. Therefore, we defined patients with increased visceral fat, despite the decreased PMI, as BCC group. These changes in body composition in esophageal squamous cell carcinoma patients with sarcopenic obesity have rarely been reported, and there are still no reports of short- or long-term outcomes in patients with these body changes.

In our study, it was not sarcopenia before and after NAC that affected perioperative complications, but BCC group, a group in which PMI decreased but visceral fat mass increased during NAC. The relationship between perioperative complications and skeletal muscle mass or sarcopenia has been previously reported. The risk of postoperative pneumonia and overall perioperative complications is increased by preoperative sarcopenia [9,35,36], which is reportedly mainly due to respiratory dysfunction associated with decreased respiratory-related muscle strength in sarcopenia patients [37]. In contrast, cases with reduced PMI during NAC were reportedly associated with postoperative mortality, although there was no association with postoperative complications [38]. Since the above reports focused on skeletal muscle mass and not on changes in visceral fat, some of the cases with reduced skeletal muscle mass may also have had increased visceral fat. A report on NAC in pancreatic cancer [39] and esophageal adenocarcinoma [28] showed that sarcopenic obesity was associated with postoperative complications in sarcopenia.

This study focused on patients with decreased PMI but increased visceral fat mass (BCC group) during NAC and found that these cases were associated with postoperative complications. Even if the amount of visceral fat mass increased before NAC, there were no cases of visceral obesity exceeding 100 cm^2^. In this group, NAC effectively relieved dyspnea and dysphagia to a certain extent, and oral intake increased, and nutritional therapy using parenteral nutrition (PN) and enteral nutrition (EN) allowed them to consume more calories than before NAC. However, they could not exercise due to the fatigue caused by NAC’s side effects, which may have caused the loss of muscle mass. Since these patients appear to have increased their dietary intake, the lack of active multidisciplinary intervention involvement by the NST may be the reason for the presence of BCC group. In patients with low muscle mass, there is a correlation between muscle mass and perioperative outcomes since they do not have sufficient reserves for protein synthesis and glycogenesis [40]. Obese patients have been shown to have a higher risk of anastomotic leakage [41]. In addition, in esophageal cancer surgery, obesity may lead to increased acid reflux, increased intra-abdominal pressure, and vagal abnormalities that promote the secretion of bile and pancreatic enzymes and increase complications [42]. Due to these risk factors, it can be concluded that BCC group was significantly associated with postoperative complications.

The OS rate was evaluated as a long-term outcome. In our study, the prognostic factors for patients with resectable thoracic esophageal cancer were stage III and post-NAC sarcopenia. However, inclusion in BCC group and pre-NAC sarcopenia were not prognostic factors. In previous reports of esophageal cancer, the loss of muscle mass during NAC was associated with OS and progression-free survival rather than sarcopenia [43]. However, the relationship with changes in body composition, including changes in visceral fat mass, was less clear. Skeletal muscle loss is thought to be caused by an imbalance between tumor-derived factors, inflammatory cytokines that inhibit protein synthesis [44], and myokines, which are cytokines produced and released by skeletal muscle [45]. These cytokines and impaired immunity due to malnutrition have been reported to affect prognosis [46]. In recent years, muscle quality, especially intermuscular adipose tissue (IMAT), has been associated with survival [47]. Recently, IMAT has been reported as a mortality risk factor for coronavirus disease 2019 [48]. It is possible that BCC group was not associated with OS in our study since OS was associated with IMAT, rather than changes in visceral fat mass, and further study is needed to elucidate this.

In this study, approximately 30% of patients had strong malignant stenoses that prevented the endoscope’s passage, forcing the introduction of IVH and consideration of dietary patterns. However, it has been reported that approximately 80% of patients with esophageal cancer are malnourished before the start of treatment, mainly due to dysphagia and anorexia. Loss of muscle mass and muscle strength have been reported without changes in weight, functional physical activity, or lifestyle during NAC [49,50], suggesting the need for nutritional support and exercise therapy. This is the case in our results, and it is necessary to intervene to prevent the loss of skeletal muscle mass during NAC. The European Society for Clinical Nutrition and Metabolism (ESPEN) guidelines [51] strongly recommend nutritional interventions, including nutritional guidance and oral nutritional supplements, to increase oral intake in patients with cancer. It has been reported that early EN rather than PN prevents muscle mass loss [52], reduces perioperative complications, and shortens hospital stay in cases of esophageal cancer in which the patient has difficulty in taking adequate oral intake due to transit obstruction [53]. The guidelines [51] state that physical activity to maintain muscle mass and physical function is necessary and strongly recommend exercise therapy. Although it is difficult to recommend exercise to all patients, the guidelines recommend exercise therapy tailored to each case to reduce inactivity and avoid a sedentary lifestyle, and physical activity should be recommended even if it starts with a daily walk. A physical exercise program by an appropriately trained professional is recommended for patients who have difficulty with independent exercise therapy. Chmelo et al. [54] reported a trial of home rehabilitation during NAC, suggesting that it may improve health status during NAC and preoperatively and prevent the deterioration of cardiopulmonary function. In order to improve prognosis and reduce perioperative complications, it is necessary to prevent loss of muscle mass not only by nutritional management but also through intervention by exercise therapy, including at-home rehabilitation.

This study had several limitations. First, this was a single-center, retrospective study conducted at our institution. The cases reviewed in this study included those who were introduced for the purpose of surgery, and there was a selection bias since many of them had no problems with operability. In addition, the number of cases was limited since this was a single-center study. Second, in our study, nutritional assessment was performed before and after NAC, but not all patients received adequate nutritional management by NST due to limited intervention during hospitalization. In fact, even in patients with severe esophageal stricture due to esophageal cancer, only oral intake was used for nutritional management, and EN was used in eight cases and PN in six cases. Nutritional supplements were provided to patients whose nutrition could not be adequately managed by diet alone, but adequate feeding was not possible due to preference issues.

## 5. Conclusions

Although NAC of esophageal cancer is a short-term treatment, changes in body composition during NAC affected perioperative complications and OS. Interventions including exercise therapy and nutritional interventions from the start of NAC can prevent loss of skeletal muscle mass and increase visceral fat mass and may improve OS and perioperative complications.

## Figures and Tables

**Figure 1 jcm-11-00508-f001:**
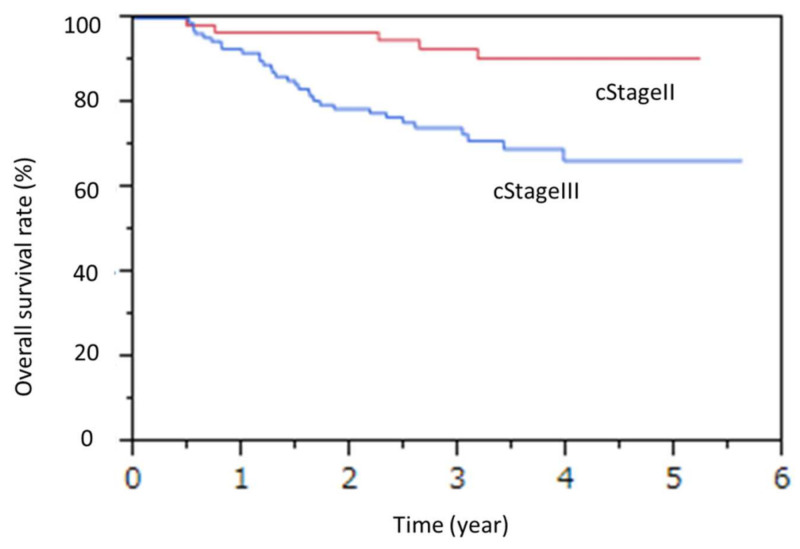
Overall survival by cStage.

**Figure 2 jcm-11-00508-f002:**
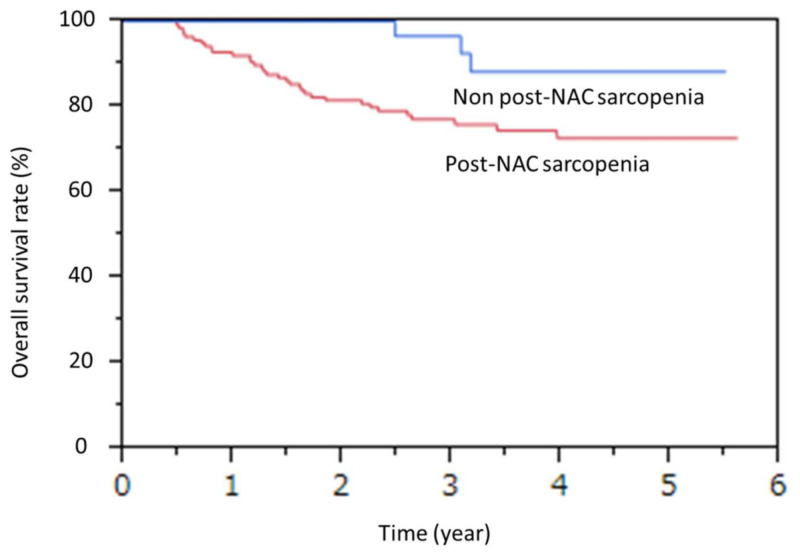
Comparison of overall survival between Post-NAC sarcopenia and Non post-NAC sarcopenia.

**Table 1 jcm-11-00508-t001:** Background factors of total patients.

	Total
*n*	175
Age (years)	66.2 ± 7.7
Sex (M/F), *n*	139/36
Diabetes mellitus, *n* (%)	24 (13.7)
Cardiovascular disease, *n* (%)	71 (40.5)
Drinking, *n* (%)	139 (79.9)
Brinkman index	606.7 ± 514.2
ASA/PS(1/2/3)	65/109/1
% FEV1	76.7 ± 7.6
% VC	100.9 ± 12.7
Weight loss rate (%)	3.2 ± 4.6
BMI (kg/m^2^)	21.1 ± 3.0
Alb (g/dL)	4.1 ± 0.4
PNI	49.3 ± 4.7
SGA (A/B/C)	135/24/16
NST intervention, *n* (%)	17 (9.7)
NAC (FP/DCF), *n*	122/53
Location (Ut/Mt/Lt), *n*	36/87/52
cStage (II/III), *n*	64/111
Side effect of chemotherapy	
Fatigue, *n* (%)	14 (8.0)
Appetite loss, *n* (%)	17 (9.7)
Febrile neutropenia, *n* (%)	91 (51.9)

M: male; F: female; ASA-PS:American Society of Anesthesiologists Physical Status; FEV1.0: Forced Experiratory Volume in one second; VC: Vital Capacity; BMI: Body Mass Index; Alb: Albumin; PNI: Prognostic Nutritional Index; SGA: Subjective Global Assessment; NST: Nutritional Support Team; NAC: Neoadjuvant Chemotherapy.

**Table 2 jcm-11-00508-t002:** Change in psoas muscle index (PMI) and visceral fat mass of 3% or more before and after neoadjuvant chemotherapy (NAC).

		Visceral Fat Mass		
		Increase	Decrease	Total
PMI	Increase	23	9	32
	Decrease	57	86	143
	Total	80	95	175

**Table 3 jcm-11-00508-t003:** Characteristics of BCC group.

	BCC Group	Non-BCC Group	*p*-Value
*n*	57	118	
Age (years)	66.3 ± 7.5	66.1 ± 7.9	0.86
Sex (M/F), *n*	40/17	99/19	0.03
Diabetes mellitus, *n* (%)	7 (12.3)	17 (14.5)	0.81
Cardiovascular disease, *n* (%)	19 (33.3)	52 (44.4)	0.18
Drinking, *n* (%)	47 (82.5)	92 (78.6)	0.69
Brinkman index	644.2 ± 566.2	588.5 ± 488.4	0.50
ASA/PS (1/2/3), *n*	26/31/0	39/78/1	0.22
% FEV1	76.9 ± 7.6	76.6 ± 7.6	0.83
% VC	104.1 ± 13.1	99.3 ± 12.2	0.71
BMI (kg/m^2^)	20.56 ± 2.9	23.1 ± 2.3	<0.01
Prealbumin (mg/dL)	25.2 ± 5.3	26.4 ± 5.8	0.16
SGA (A/B/C), *n*	40/11/6	95/13/10	0.27
NST care, *n* (%)	7 (12.3)	10 (8.5)	0.42
NAC (FP/DCF), *n*	41/16	81/37	0.72
Location (Ut/Mt/Lt), *n*	12/30/15	24/57/37	0.78
cStage (II/III), *n*	19/38	45/73	0.61
Side effect of chemotherapy			
Fatigue, *n* (%)	6 (10.5)	8 (6.8)	0.33
Appetite loss, *n* (%)	7 (12.3)	10 (8.5)	0.39
Febrile neutropenia, *n* (%)	32 (55.0)	59 (49.9)	0.86
Diarrhea, *n* (%)	4 (7.0)	3 (2.5)	0.50
Therapeutic effect (PR/SD)	47/10	94/23	0.83

**Table 4 jcm-11-00508-t004:** Body composition and nutrition parameters pre- and post-NAC in BCC group.

	BCC Group	Non-BCC Group	*p*-Value
*n*	57	118	
Pre-NAC sarcopenia, *n* (%)	44 (77.2)	95 (80.5)	0.69
Post-NAC sarcopenia, *n* (%)	45 (78.9)	95 (80.5)	0.85
Pre-NAC visceral fat mass (cm^2^)	79.7 ± 55.6	88.1 ± 48.8	0.30
Post-NAC visceral fat mass (cm^2^)	92.5 ± 58.5	77.2 ± 42.6	0.05
Pre-NAC alb (g/dL)	4.1 ± 0.3	4.1 ± 0.4	0.65
Post-NAC alb (g/dL)	3.9 ± 0.3	3.9 ± 0.3	0.96
Pre-NAC PNI	49.7 ± 4.5	49.2 ± 4.7	0.46
Post-NAC PNI	47.3 ± 5.0	46.6 ± 4.0	0.39

**Table 5 jcm-11-00508-t005:** Comparison of perioperative complications, operative time, and hospital stay in pre- and post-NAC sarcopenia, and BCC group.

	Pre-NAC Sarcopenia	Non-Pre-NAC Sarcopenia	*p*-value
	(*n* = 139)	(*n* = 36)	
Total Clavien Dindo > 3, *n* (%)	44 (31.7)	13 (36.1)	0.69
Length of Hospitalization(day)	24 ± 13.9	20.2 ± 8.9	0.12
Operation Time (min)	467.4 ± 87.1	464.3 ± 85.3	0.84
	Post-NAC sarcopenia	Non-post-NAC sarcopenia	*p*-value
	(*n* = 140)	(*n* = 35)	
Total Clavien Dindo > 3, *n* (%)	44 (31.4)	13 (37.1)	0.54
Length of Hospitalization(day)	24.1 ± 13.8	19.8 ± 8.8	0.07
Operation Time (min)	465.5 ± 86.5	471.5 ± 87.5	0.71
	BCC Group	Non-BCC	*p*-value
	(*n* = 57)	(*n* = 118)	
Total Clavien Dindo > 3, *n* (%)	27 (47.3)	30 (25.4)	<0.01
Length of Hospitalization (day)	26.3 ± 15.1	21.7 ± 11.8	0.03
Operation Time (min)	460.9 ± 71.8	469.6 ± 92.9	0.53

**Table 6 jcm-11-00508-t006:** Cox proportional hazard model of overall survival (OS) using univariate and multivariate analyses.

	Univariate Analysis	Multivariate Analysis
HR	*p*-Value	HR	*p*-Value
Age	≤70 years	1.05 (0.54–2.13)	0.88		
	>70 years	1			
Gender	M	1.25 (0.56–3.33)	0.6		
	F	1			
NAC	DCF	1.08 (0.51–2.15)	0.82		
	FP	1			
△PMI	down	1.09 (0.44–2.36)	0.82		
	up	1			
△Visceral Fat	up	1.38 (0.71–2.77)	0.33		
	down	1			
cStage	III	4.15 (1.76–12.2)	<0.01	3.94 (1.67–11.58)	<0.01
	II	1		1	
Diabetes Mellitus	+	2.23 (0.98–4.58)	0.06		
	-	1			
Pre-sarcopenia	+	2.3 (0.92–7.8)	0.07		
	-	1			
Post-sarcopenia	+	3.23 (1.15–13.4)	0.02	2.92 (1.04–12.1)	0.03
	-	1		1	
BCC Group	+	1.32 (0.66–2.89)	0.43		
	-	1			
Therapeutic Effect	≤SD	1.82 (0.83–3.6)	0.12		
	≥PR	1

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
