# Peer review of "Effect of Body Composition Change during Neoadjuvant Chemotherapy for Esophageal Squamous Cell Carcinoma"

_jcm, 2022, doi:10.3390/jcm11030508_

Round 1

Reviewer 1 Report

I congrats to the authors for the interisting work! Some general language revisions should be made  as the introduction is not very clear in the present form. 

Author Response

I congrats to the authors for the interisting work! Some general language revisions should be made as the introduction is not very clear in the present form.

Thank you for pointing this out. This paper has been proofread by EDITAGE for English editing. We will review it again.

Reviewer 2 Report

Onishi et al. present a study about the change of body composition in patients with esophageal squamous cell carcinoma undergoing neoadjuvant chemotherapy.

I have several concerns.

In the introduction it should be mentioned, that ESCC is not equally distributed worldwide, as the authors pointed out, that ESCC is the predominant entity for esophageal cancer worldwide. The authors should introduce the used chemotherapy regimes in more detail and might reflect the differences caused by the different regimes.  

The last sentence is wrong: “This study investigated the effects of body composition changes before and after NAC on perioperative complications and prognosis in patients with stage II or III resectable thoracic esophageal cancer.” There is no causal clue. It must be like this: The study investigated the effect of NAC to the body composition in stage II and III resectable ESCC patients.

Did all 177 patients receive the same chemotherapy? I suggest to give “Group A” a name, which is more related to the features of this group, as there is also no group B.

Numbers in Table 5 do not reflect 177 patients. The OS of the patients should be shown, not only the HR. The OS of the patients reflects the quality of therapy in a high volume center.

Why both cohorts, Group A and Others, show a sarcopenia after NAC (approx. 80%). The decreased OS in sacropenic patients is independent, whether they are in Group A or Others?

I cannot underline this statement: “This study focused on a relatively large number of patients with decreased PMI but increased visceral fat mass (group A) during NAC and found that these cases were associated with postoperative complications.”  As the group is not large (n=57). Taken together, it is well know, that patients with a poor nutritional status, have lower outcomes in their post-operative curse and also in their overall survival.

The recent literature should be considered: https://pubmed.ncbi.nlm.nih.gov/?term=esophageal+cancer+sarcopenia+&sort=pubdate

Author Response

In the introduction it should be mentioned, that ESCC is not equally distributed worldwide, as the authors pointed out, that ESCC is the predominant entity for esophageal cancer worldwide. The authors should introduce the used chemotherapy regimes in more detail and might reflect the differences caused by the different regimes. 

Thank you for pointing this out. I have added the regional differences to l5-8 in the introduction. Chemotherapy is described in 2.2 (Neoadjuvant chemotherapy) on page 2. Chemotherapy is administered to all patients. The chemotherapy regimen is FP or DCF, and the treatment regimen is decided at the conference. In this study, as shown in Table 4 and Table 7, there was no difference by NAC regimen.

The last sentence is wrong: “This study investigated the effects of body composition changes before and after NAC on perioperative complications and prognosis in patients with stage II or III resectable thoracic esophageal cancer.” There is no causal clue. It must be like this: The study investigated the effect of NAC to the body composition in stage II and III resectable ESCC patients.

Thank you for pointing this out.

We have made the changes as you suggested.

Did all 177 patients receive the same chemotherapy?

Thanks for the question.

As stated in P3, 3.1(Background factor), chemotherapy is given in FP and DCF, and NAC was performed in FP/DCF = 122/53 of patients.

I suggest to give “Group A” a name, which is more related to the features of this group, as there is also no group B.

Thank you for pointing this out.

As you pointed out, it was difficult to understand, so we have changed ‘GroupA’ to ‘Body composition change (BCC) group’.

Numbers in Table 5 do not reflect 177 patients.

Thank you for pointing this out.

There were a total of 175 cases in this study. This was a mistake and has been corrected.

The OS of the patients should be shown, not only the HR. The OS of the patients reflects the quality of therapy in a high volume center.

Thanks for the suggestion, I have added the OS to P7 3.7(prognostic factor for OS). In addition, a figure on overall survival has been added to the Supplementary Materials.

And as you have pointed out, we are a high volume center and therefore have a large number of cases. But we do treat with standard treatment.

Why both cohorts, Group A and Others, show a sarcopenia after NAC (approx. 80%). The decreased OS in sacropenic patients is independent, whether they are in Group A or Others?

Thank you for pointing this out.

In this study, if the PMI is below the definition, it is considered sarcopenia. Since the BCC group is divided by the rate of change, both groups will include post-NAC sarcopenia. OS is also significantly lower in post-NAC sarcopenia, independent of the change group.

I cannot underline this statement: “This study focused on a relatively large number of patients with decreased PMI but increased visceral fat mass (group A) during NAC and found that these cases were associated with postoperative complications.”  As the group is not large (n=57). Taken together, it is well know, that patients with a poor nutritional status, have lower outcomes in their post-operative curse and also in their overall survival.

Thank you for pointing this out.

We focused on patients who had an increase in visceral fat mass in conjunction with a decrease in skeletal muscle mass. Although there have been reports of sarcopenic obesity, there have been no reports of the BCC group that we reported. Therefore, we used the expression "relatively large number of patients," but as you pointed out, it was about 33%, so we deleted the part you pointed out.

As you mentioned, in general, malnutrition is associated with many postoperative complications and a poor prognosis. However, during the course of treatment, there were patients whose skeletal muscle mass (muscle mass) had decreased, even though their nutritional status seemed to have improved from clinical and hematological findings. In our study, we believe that the decrease in skeletal muscle mass during NAC leads to poor results. Therefore, we think it is necessary to consider the combination of rehabilitation that is not only nutritional management.

The recent literature should be considered: https://pubmed.ncbi.nlm.nih.gov/?term=esophageal+cancer+sarcopenia+&sort=pubdate

Thank you very much. I referred to the recent literature on esophageal adenocarcinoma (28) from the literature you provided. This literature evaluates the prevalence of post-NAC sarcopenia and post-NAC sarcopenic obesity perioperative body composition changes using CT. And they were examining the effects of body composition changes on postoperative complications and overall survival. They reported that post-NAC sarcopenia in esophageal adenocarcinoma was associated with significantly more postoperative complications and a trend toward a worse prognosis. However, we are the first to report the effects of body composition changes during NAC in esophageal squamous cell carcinoma on their postoperative complications and OS. In particular, there are few reports that evaluate the BCC group, which has decreased skeletal muscle mass and increased visceral fat mass.

Reviewer 3 Report

I'd like to congratulate the authors on a well-written article and interesting study.

I have no major comments about the methodology, results, discussions, or conclusions.

My only remark concerns the retrospective nature of the study. It is often burdened with selection bias. In the case of the analyzed work, it does not seem to me to have a significant impact on the quality of the results.

I wish the authors good luck in their further research and personal life. 

Author Response

I'd like to congratulate the authors on a well-written article and interesting study.

I have no major comments about the methodology, results, discussions, or conclusions.

My only remark concerns the retrospective nature of the study. It is often burdened with selection bias. In the case of the analyzed work, it does not seem to me to have a significant impact on the quality of the results.

I wish the authors good luck in their further research and personal life.

Thank you for your advice. I will refer to it for future papers.

Round 2

Reviewer 2 Report

The authors answered all questions and comments sufficiently.